# AvP: A software package for automatic phylogenetic detection of candidate horizontal gene transfers

**Georgios D. Koutsovoulos** *, **Solène Granjeon Noriot**, **Marc Bailly-Bechet**, **Etienne G. J. Danchin**, **Corinne Rancurel**

Institut Sophia Agrobiotech, Université Côte d'Azur, INRAE, CNRS, Sophia Antipolis, France

* gdkoutsovoulos@gmail.com

**Data Availability Statement:** All relevant data are within the manuscript.

**Funding:** This work has been supported by the French government, through the UCA-JEDI

## Abstract

Horizontal gene transfer (HGT) is the transfer of genes between species outside the transmission from parent to offspring. Due to their impact on the genome and biology of various species, HGTs have gained broader attention, but high-throughput methods to robustly identify them are lacking. One rapid method to identify HGT candidates is to calculate the difference in similarity between the most similar gene in closely related species and the most similar gene in distantly related species. Although metrics on similarity associated with taxonomic information can rapidly detect putative HGTs, these methods are hampered by false positives that are difficult to track. Furthermore, they do not inform on the evolutionary trajectory and events such as duplications. Hence, phylogenetic analysis is necessary to confirm HGT candidates and provide a more comprehensive view of their origin and evolutionary history. However, phylogenetic reconstruction requires several time-consuming manual steps to retrieve the homologous sequences, produce a multiple alignment, construct the phylogeny and analyze the topology to assess whether it supports the HGT hypothesis. Here, we present *AvP* which automatically performs all these steps and detects candidate HGTs within a phylogenetic framework.

This is a *PLOS Computational Biology* Software paper.

## Introduction

The acquisition of genes through horizontal gene transfer (HGT) is mostly observed in prokaryotes, where they play a significant role in adaptive evolution (e.g. antibiotic resistance). To a lesser degree, cases of HGT have also been observed in eukaryotes with important consequences in the biology of the organism [1]. The increase of new genomes being sequenced and the prediction of new gene sets, represents an opportunity to detect additional HGT cases and

"Investments in the Future" project managed by the National Research Agency (ANR) with the reference number ANR-15-IDEX-01. GDK has received the support of the EU in the framework of the Marie-Curie FP7 COFUND People Programme, through the award of an AgreenSkills+ fellowship (under grant number 609398).

to characterize more precisely the possible donors. To sustain these needs, high-throughput yet robust HGT detection methods are required.

One method to predict potential HGTs is to calculate the difference in similarity using BLAST [2] (or other sequence similarity search software) between phylogenetically closely related and distant species. The Alien Index (AI) metric uses the difference in e-value between the best hit from closely (Ingroup) and distantly (Donor) related taxa [3]. Positive AI means that the gene is more similar to a distant taxon and indicates a potential HGT. In the past, different values of AI have been used as a cutoff to decrease false positives but with the potential risk of missing HGTs. Similarly, the HGT Index (h) [4] uses the difference in bit scores but is hampered by the same limitations in terms of a trade-off between reducing false positives without missing valid cases. However, tracking these false positives from homology search results alone is not possible.

Even if different cutoffs are applied to AI, the underlying best BLAST hit analysis is an over-simplistic method for the evolutionary complexity of HGT. Recently, an additional metric called outg_pct, which is the percentage of species from Donor lineage in the top hits that have different taxonomic species names, has been used in conjunction with AI to filter out some of the false positives resulting from erroneous taxonomic annotation of the best blast hits [5]. A more evolutionary comprehensive method is to extract the results from the whole BLAST analysis and infer a phylogenetic tree. The phylogenetic position of the potential HGT candidate in relation to the other genes and their taxonomy will provide an evolutionary framework and will validate or reject the HGT hypothesis. However, manually producing then checking each phylogenetic tree is a labour-intensive and time-consuming process. In addition, contamination or symbionts in genome sequencing, unless handled properly, can provide false positives that pass both AI and phylogenetic analysis [6]. External information, such as the target gene structure, taxonomic affiliation of genes near the target gene, and support by transcription data are necessary to eliminate such false positives. Combining all information will lead to a more accurate prediction of putative HGTs.

Methods exist to perform gene tree species tree reconciliation to detect xenologs (i.e HGTs) [7, 8] and are able to distinguish genes that were transferred horizontally with or without duplication events. However, providing a species tree together with the gene tree is required. Therefore, testing hundreds of genes requires either creating different species trees according to the input sequences or comparing everything against the whole NCBI tree of life containing hundreds of thousands of branches.

In this study, we present *AvP* (short for 'Alienness vs Predictor') to automate the robust identification of HGTs at high-throughput with no need to provide a reference species tree. *AvP* extracts all the information needed to produce input files to perform phylogenetic reconstruction, evaluate HGTs from the phylogenetic trees, and combine multiple other external information for additional support (e.g. gff3 annotation file, transcript quantification file). Our method does not rely on an explicit reference species tree and only uses a simplified take on the species phylogeny, according to the organism tested. This allows for a rapid phylogenetic detection of HGTs that can then be used as input for more sophisticated analyses.

## Design and implementation

### Software description

*AvP* performs automatic detection of HGT candidates within a phylogenetic framework. The pipeline comprises two major steps: (i) *prepare*, and (ii) *detect*, and three optional steps: (iii) *classify*, (iv) *evaluate*, and (v) *hgt_local_score* (Fig 1). Although the pipeline has been

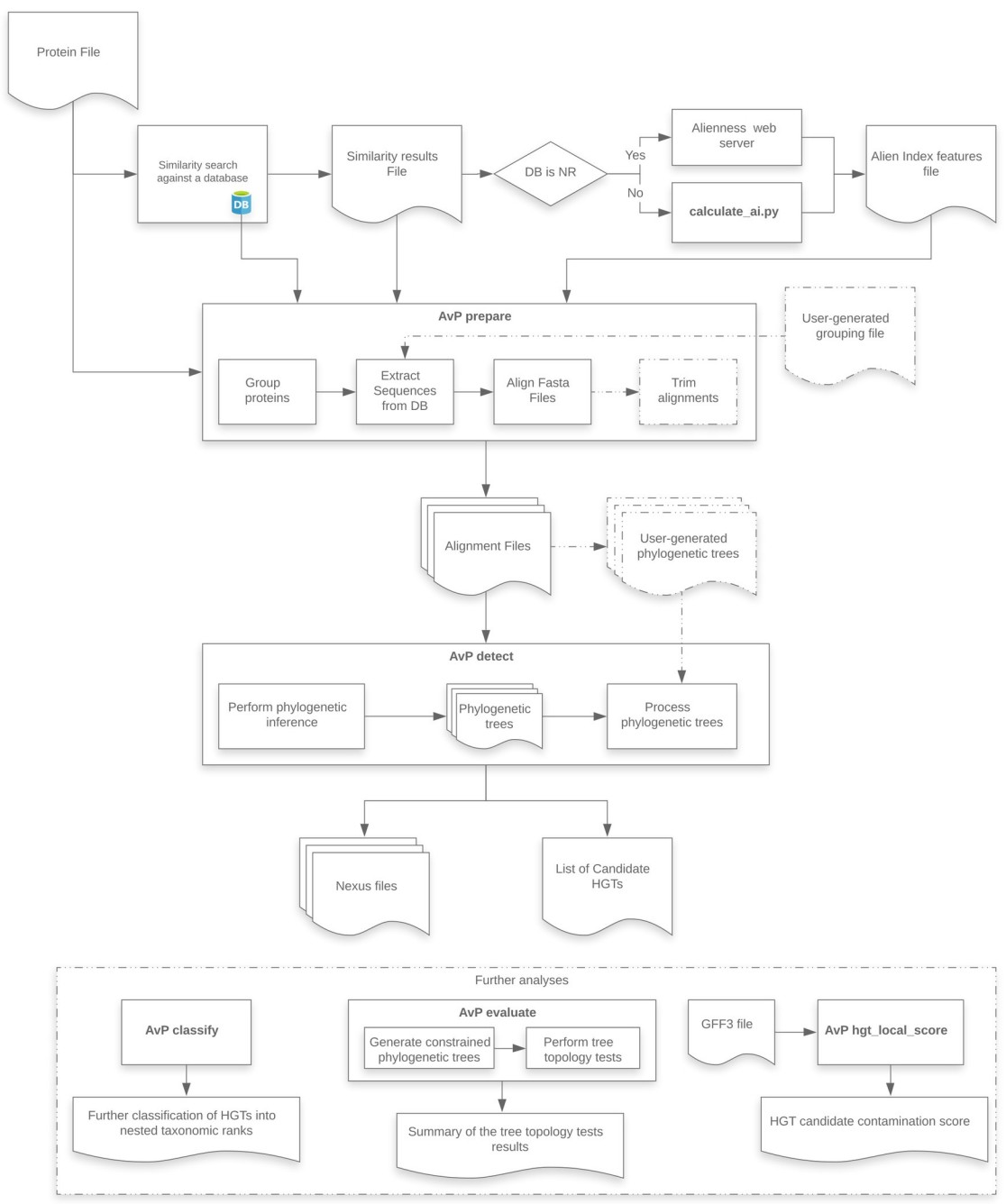

**Fig 1. AvP workflow.** Dashed lines indicate optional routes and analyses.

extensively tested with protein datasets, it should work at the DNA level, including non-coding sequences (see GitHub documentation). For the rest of the article we assume a protein dataset.

*AvP* requires three primary files, (i) a fasta file containing the proteins of the species being studied, (ii) a tabular results file of similarity search (e.g. BLAST or DIAMOND [9]) against a protein database, and (iii) an AI features file. Furthermore, the user must provide two config files, one with information on the taxonomic ingroup in the study (defining which group of

species is considered closely related and which group is distantly related) and one defining multiple software parameters. The AI features file can be created with the script *calculate_ai.py* which can be found in the repository.

## AvP prepare

The software selects proteins for downstream analyses based on any combination of the metrics AI, outg_pct, and AHS (described below). Then, the software collects all protein sequences corresponding to significant hits from the database based on the tabular results file of the homology search and groups the query species sequences based on the percentage of shared hits (by default 70%) using single linkage clustering. Alternatively, the user can specify a file containing user-generated groups of queries and hits (e.g. from OrthoFinder [10] or protein domain analysis). For each group, a fasta file is created containing the query species sequences and their respective database hits. Each file is then aligned using MAFFT [11] with an option for alignment trimming with trimAl [12].

## AvP detect

There are two options available for phylogenetic inference within *AvP*: (i) FastTree [13], and (ii) IQ-TREE [14]. The defaults for these programs are [-gamma -lg] for FastTree and [-mset WAG,LG,JTT -AICc -mrate E,I,G,R] for IQ-TREE. The user can change the IQ-TREE parameters in the config file. These two approaches vary in time and compute requirements, and consequently in tree reconstruction accuracy [15]. Alternatively, the software can utilise user-generated phylogenetic trees using the alignment files created with *AvP prepare* with any program that can produce a valid Newick tree format file. By default, *AvP* does not impose a branch support threshold. However, the user can define a support threshold in the config file under which branches collapse into polytomies.

Each phylogenetic tree is then processed (midpoint rooting) and each query sequence is classified into one of the following three categories: HGT candidate (✓), Complex topology (?), No evidence for HGT (X). The taxonomic assignment of genes and their position in the tree relative to the query gene are used to characterise the gene as HGT or not. Two branches are taken into account, the sister branch of the gene of interest and the ancestral sister branch (Fig 2). Both of these branches are tagged independently depending on the included sequences to either Donor (i.e distantly related species), Ingroup (i.e closely related species), or both. Ingroup is defined by the user and Donor is all species not in Ingroup. The Ingroup tag is applied if most of the sequences (default 80%) belong to taxa inside the taxonomic group closely related to the species studied. Consequently, the Donor tag is applied if most of the sequences belong to taxa that fall outside of the Ingroup taxonomic clade. If the branch contains taxa from both groups at a ratio higher than 1 to 5, then the branch is tagged as both. The tags of these two branches are then processed according to Table 1. For example, if we are searching in a eukaryotic species for HGT originating from prokaryotic species, the Ingroup is set to Eukaryota and the Donor to non Eukaryota (bacteria, viruses etc). If the sister branch of the query contains sequences that belong to eukaryotic species, it is tagged as Ingroup and the gene is not considered as an HGT. In another example, if both the sister branch and the ancestral sister branch contain mostly sequences from non eukaryotic species, both of the branches are tagged as Donor ant the gene is considered as a potential HGT.

For each query sequence, the software produces a nexus formatted file containing the phylogenetic tree, the taxonomic information for each sequence, and each sequence coloured by the taxonomic affiliation for quick visual parsing. The nexus file can be visualised with the tree visualisation software FigTree [16].

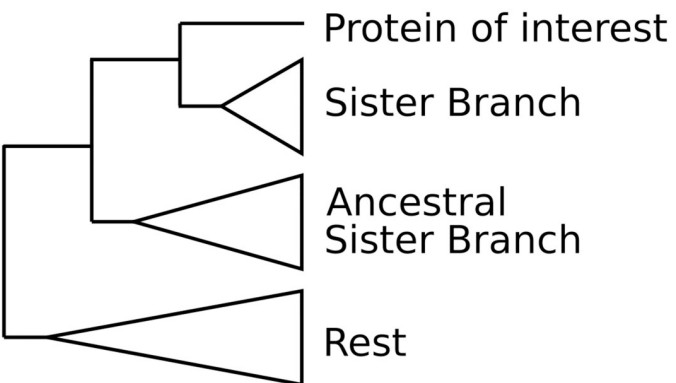

**Fig 2. Tree example.** Sister branch positions on the phylogenetic tree.

## AvP classify

This step allows the further classification of HGT candidates into user-generated nested taxonomic ranks for their putative origins. It follows the same logic as in the step *AvP detect* described previously in terms of tagging the clades to a specific taxonomic affiliation. For example, the HGTs can be classified based on their origin, such as Fungi, Viridiplantae, Viruses etc., according to the NCBI taxonomy.

## AvP evaluate

For each HGT candidate, the topology is constrained to form a single monophyletic group containing the query sequence and all the Ingroup sequences. A phylogenetic tree is inferred with FastTree or IQ-TREE and the likelihoods of the initial and constrained topologies are compared with IQ-TREE, which supports several tree topology tests. This step can inform whether the topology supporting HGT is more likely than the alternative constrained topology that does not support HGT.

## AvP hgt_local_score

Given a gff3 file containing the genomic location of the genes of the query species and the results of the *AvP* analyses, this step calculates a score for each HGT candidate that corresponds to whether the HGT candidate is surrounded by genes from the query genome or 'alien' genes, including possible contaminants. The score ranges between -1 and +1, with -1 indicating strongly a contamination while +1 indicating strongly a HGT candidate (Fig 3). The rationale is that a candidate HGT surrounded by genes that were also detected as candidate HGT might be part of a contaminant insertion in the genome assembly (although HGT of a whole block of genes or duplications after acquisition are also possible). Hence, this step

**Table 1. Detection table whether the gene tested is an HGT candidate.**

| Ancestral SB | Sister branch (SB) | | |
|---|---|---|---|
| | **Donor** | **Ingroup** | **Donor + Ingroup** |
| Donor | ✓ | X | ? |
| Ingroup | ? | X | X |
| Donor + Ingroup | ? | X | ? |
| Not present | ✓ | X | ? |

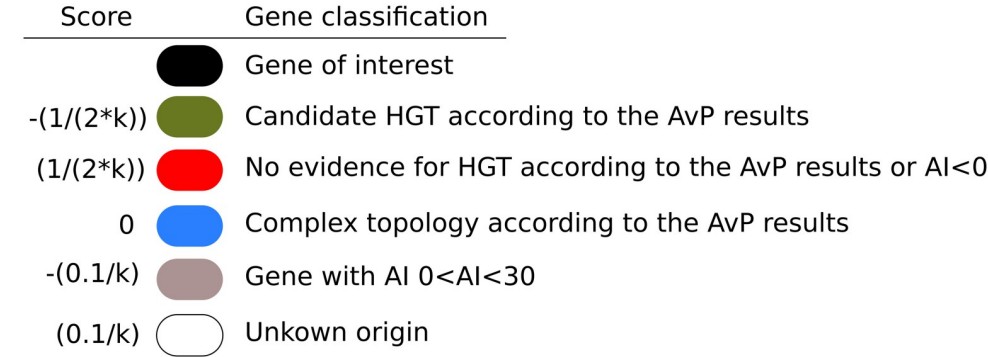

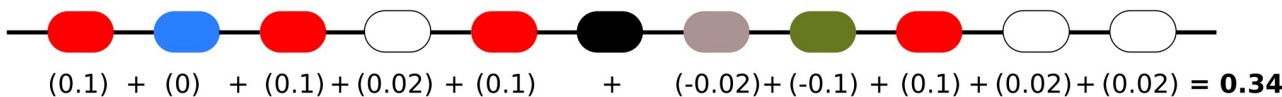

**Fig 3. HGT local score calculation.** Each neighbouring gene contributes to the score based on its classification getting a value described in the top left panel. In the example, the score is equal to 0.34, most likely indicating an HGT insertion. Overall, a score above 0 indicates an HGT insertion, while a score below 0 indicates a possible contamination or HGT rich region.

allows alerting the user on possible contaminations. On the opposite, if the candidate HGT is surrounded by genes that were more likely inherited vertically, the contamination hypothesis can be reasonably ruled out.

## AHS: A new contamination-aware metric

The two metrics that are widely used (AI and h) utilise only the best Ingroup and Donor hit from the BLAST output. This poses several potential issues and AI in particular can be 0 if both hits have e-value of 0, although they can differ in similarity. The h metric resolves this problem by using the bitscore instead of the e-value. However, both of these metrics are sensitive to taxonomically misclassified or contaminating sequences in databases as they only rely on the best hits. For instance, if the best hit is wrongly assigned a Donor taxid in the database, these metrics will erroneously detect a candidate HGT. In the opposite, a wrongly assigned Ingroup taxid in the database would necessarily result in no HGT detection if it is the best hit. A different approach to try to circumvent this issue is to aggregate all the bitscores of Donor and Ingroup sequences and then perform the calculation of h. Although this approach will minimise erroneous results it will still suffer from sampling biases.

In order to minimise all these effects we developed a new metric called Aggregate Hit Support (AHS). We first normalise each bitscore Eq (1). We then sum all the normalised bitscores of the Donor hits and all the normalised bitscores of the Ingroup hits seperately and calculate the difference Eq (2). A positive AHS score suggests a potential HGT candidate.

$$Bitscore_N = Bitscore \cdot e^{-10 \cdot \frac{HBitscore - Bitscore}{Bitscore}} \tag{1}$$

$$AHS = \sum Bitscore_N^{Donor} - \sum Bitscore_N^{Ingroup} \tag{2}$$

## Results

### HGT pipeline

We tested our pipeline using the predicted protein set for the tardigrade species *Hypsibius exemplaris* (previously named *H*. dujardini) [17]. We used the database NCBI nr instead of SwissProt+TrEMBL libraries, used in the original publication, and selected candidates with $AI > 30$ instead of $h_{ST} > 30$ (HGT Index), while the phylogenetic inference was performed with FastTree instead of RAxML [18]. The final selection was 401 proteins (386 genes) compared to 463 proteins (463 genes), and based on the phylogenetic trees, we detected a total of 379 candidate HGTs (95%) instead of 357 (77%). Overall, 342 candidate HGTs were common to *AvP* and the previously published analysis, the ones not identified by our pipeline having an AI below 30. We then evaluated the candidate HGTs by comparing the likelihoods of the original HGT-supporting trees to those of constrained trees in which tardigrade and other metazoan proteins were forced to form a monophyletic group. Equally likely topologies were observed for 27 proteins bringing the total number of strongly supported candidate HGTs to 352 (1.7% of the total proteins present in the genome). To assess the effect of using different databases, we performed two more searches against SwissProt (SP) and Uniref90 (UR). A total of 196 / 333 / 401 proteins were selected when using SP / UR / NR resulting in 127 / 292 / 352 candidate HGTs after alternative topology tests (*AvP evaluate*). Hence, depending on the sampling of the sequence diversity present in the sequence database, the number of detectable candidate HGT varies considerably.

In the original publication describing Alien Index (AI) [3], the authors considered $AI > 45$ to be a good indication of foreign origin while genes with $0 < AI < 45$ were designated intermediate. However, this AI threshold value was originally defined on one single species only, the bdelloid rotifer, and further analyses on plant-parasitic nematodes have shown that an $AI > 45$ might be too stringent, leaving several true positives undetectable [19]. Here, we calculated the F1 score Eq (3) for all N with $AI > 0$ in *H. exemplaris* to decide the optimal threshold between precision and sensitivity. We found that selecting genes with $AI > 10$ represented an optimal balance between sensitivity and precision (Fig 4). Therefore, we propose to perform *AvP* with $AI > 0$ with FastTree option to minimize the risk of missing HGT cases and utilise the scripts provided to calculate the F1 score and based on that, decide the optimal AI threshold (which is 10 for tardigrade example) for more sophisticated and time-consuming phylogenetic analyses.

$$F1_N = 2 \cdot \frac{HGT_{AI>N}}{HGT_{AI>0} + Genes_{AI>N}} \tag{3}$$

### AHS metric

To test the new metric, we performed an *AvP* analysis with the nematode *Caenorhabditis elegans* and excluding members of the Nematoda phylum from the metazoan matches. Hence, the analysis was configured to identify HGT of non-metazoan origin in *C. elegans* and possibly present in any other nematoda species. We compared the list of potential HGTs by Crisp et al., [20] with the results obtained by *AvP* (see S1 File for full results of the comparison). By using an initial filter of $AI > 0$ or $AHS > 0$ we managed to recover 5 cases that would have been missed if there were filtered only on AI.

We thoroughly checked two of these cases where AI and AHS disagreed to identify the cause. In the first case, AI is 318, indicating a strong HGT candidate, while AHS is -44029 indicating the opposite. The two proteins that are identified as Bacteria, and thus as best non-

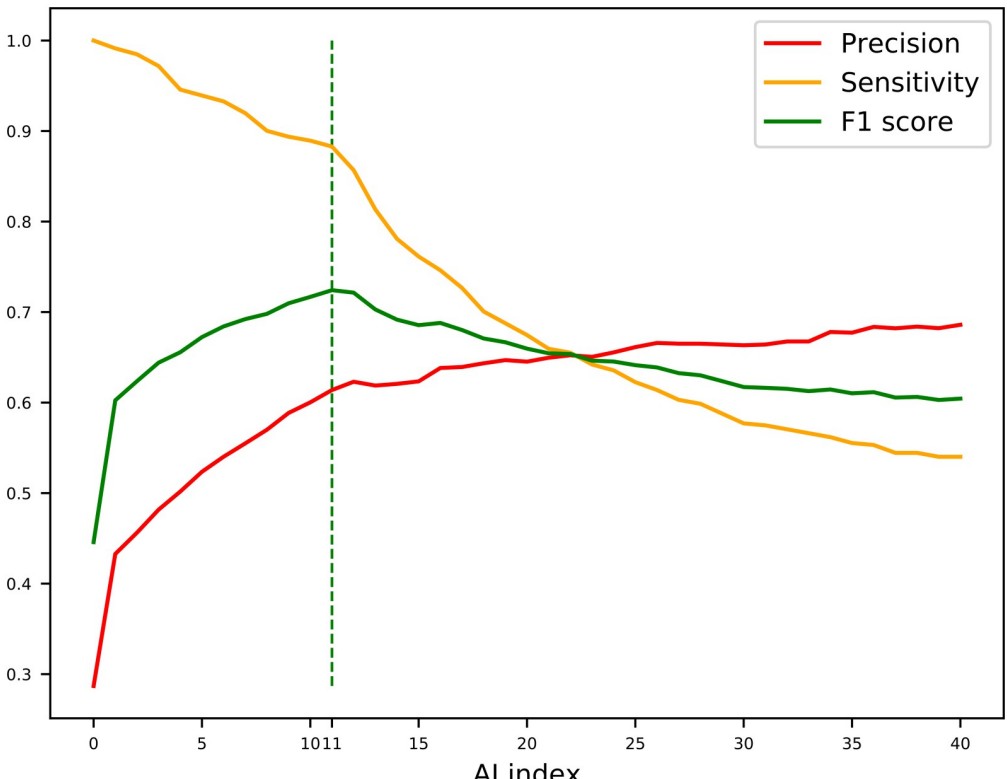

**Fig 4. Sensitivity, precision, and F1 score calculations.** Sensitivity, Precision, and F1 Score were calculated for Alien Index (AI) up to 40 for the proteins of the tardigrade *Hypsibius exemplaris*. The dashed line indicates the AI with the highest F1 score indicating the most accurate AI threshold.

metazoan hits, are most likely taxonomically missclasified since they are almost identical to the nematode protein and nested in a branch otherwise containing only nematode sequences and not found in any other bacterium (S1 Fig). An alternative less likely hypothesis is that these proteins represent a recent transfer from nematodes to bacteria. In any case, this does not represent HGT from bacteria to nematodes and the AHS metric is not misled by this likely erroneous taxonomic annotation.

In the second case, AI is 7 indicating a poorly supported HGT candidate, while AHS is 10356 indicating a strong HGT candidate (S2 Fig). The closest non nematode metazoan hit is annotated as a rodent protein. Running a BLAST for this protein in nr shows that it is very similar to a nematode of the genus *Trichuris* which some of its members are shown to be rodent parasites. Thus the rodent protein is actually more likely to represent contamination from a nematode one and should have been excluded from calculating AI and AHS. Although for *C. elegans* the difference in AI and AHS appear small, performing AI and AHS calculation on the *Trichuris suis* protein results in $AI < -50$ while $AHS > 10000$, further indicating that AHS is much less sensitive to taxonomic annotation errors than AI.

Consequently, it seems that this new AHS metric is able to correct errors due to contamination and taxonomic assignation bias. We thus implemented this new metric in *AvP* and recommend to use it in combination with AI or other metric.

## Future directions and availability

We propose *AvP* to facilitate the identification and evaluation of candidate HGTs in sequenced genomes across multiple branches of the tree of life. The most common methods used so far have been based on the difference of similarity between Donor and Ingroup sequences. We also propose and implemented AHS, a new metric aiming to address contamination and erroneous taxonomic annotation. Performing phylogenetic reconstruction and alternative topology evaluation creates a framework under which more robust HGT analyses can be performed. *AvP* can contribute to rapidly populate a reliable dataset with phylogenetically supported HGT cases across the tree of life. This could eventually be used in machine learning approaches in the future attempt to predict HGT events from sequence feature themselves. Furthermore, calculating the hgt_local_score can help identify contamination and HGT hot spots in the genome. In the future, we aim to incorporate a basic module of *AvP* to the Alieness webserver [19], to facilitate usage of *AvP* for biologists not familiar with command line software.

The *AvP* software is available at https://github.com/GDKO/AvP. It is released under GNU General Public License v3.0.

## Supporting information

**S1 File. Comparing *AvP* results on *C. elegans* to Crisp et al., 2015.**
(PDF)

**S1 Fig. Phylogenetic tree for protein F40E10.3.** Nematoda proteins are excluded from the analysis (dark orange). Bacteria proteins are coloured green while Metazoan proteins are coloured orange. The two bacterial proteins returning the best non-metazoan hits belong to *Escherichia coli* and *Nitriliruptoraceae bacterium* and are almost identical to the protein from *C. elegans* indicating that they are missclasified.
(PDF)

**S2 Fig. Phylogenetic tree for protein F44B9.9.** Nematoda proteins are excluded from the analysis (dark orange). Fungal proteins are coloured light green, Metazoan proteins are coloured orange, Viridiplantae proteins are coloured teal, and other non-metazoan eukaryotic proteins are coloured blue. The best Metazoan hit (excluding nematode proteins) marked with the arrow most likely belongs to a nematode from *Trichuris* genus.
(PDF)

## Acknowledgments

We are grateful to the genotoul bioinformatics platform Toulouse Occitanie (Bioinfo Genotoul, doi: 10.15454/1.5572369328961167E12) for providing computing resources. We are also grateful to the OPAL infrastructure from Université Côte d'Azur and to the Université Côte d'Azur's Center for High-Performance Computing for providing resources and support.

## Author Contributions

**Conceptualization:** Georgios D. Koutsovoulos, Etienne G. J. Danchin, Corinne Rancurel.

**Data curation:** Georgios D. Koutsovoulos.

**Formal analysis:** Georgios D. Koutsovoulos, Etienne G. J. Danchin, Corinne Rancurel.

**Funding acquisition:** Georgios D. Koutsovoulos, Etienne G. J. Danchin.

**Methodology:** Georgios D. Koutsovoulos, Marc Bailly-Bechet, Etienne G. J. Danchin, Corinne Rancurel.

**Software:** Georgios D. Koutsovoulos, Solène Granjeon Noriot.

**Validation:** Georgios D. Koutsovoulos.

**Writing – original draft:** Georgios D. Koutsovoulos, Marc Bailly-Bechet, Etienne G. J. Danchin, Corinne Rancurel.

**Writing – review & editing:** Georgios D. Koutsovoulos, Etienne G. J. Danchin.

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
