## [Decision Letter · Decision Letter 0]

25 Aug 2022

Dear Dr Koutsovoulos,

Thank you very much for submitting your manuscript "AvP: a software package for automatic phylogenetic detection of candidate horizontal gene transfers." for consideration at PLOS Computational Biology.

As with all papers reviewed by the journal, your manuscript was reviewed by members of the editorial board and by several independent reviewers. In light of the reviews (below this email), we would like to invite the resubmission of a significantly-revised version that takes into account the reviewers' comments.

We cannot make any decision about publication until we have seen the revised manuscript and your response to the reviewers' comments. Your revised manuscript is also likely to be sent to reviewers for further evaluation.

Sincerely,

Mark Ziemann

Academic Editor

PLOS Computational Biology

William Noble

Section Editor

PLOS Computational Biology

Dear Dr Koutsovoulos,

The manuscript has been reviewed by two experts in the field and overall impressions of the software and manuscript were positive, although both reviewers requested greater clarity in the writing and explanation of certain points. Reviewer 2 point 2 has requested the software to detect non-protein coding genes, which is an excellent suggestion and of reat value to the field, however I am mindful that this may require extensive retooling of the software. I would like to flag this software feature as optional for the authors to address, however it must be mentioned as a limitation/future direction in the discussion of the manuscript along with any challenges that may be apparent when detecting HGT of non-coding genes. I look forward to receiving a revised version of the manuscript.

Regards,

Mark Ziemann, PhD

Reviewer's Responses to Questions

**Comments to the Authors:**

Reviewer #1: Koutsovolous et al. present AvP, a pipeline to automatically identify horizontally transferred genes, along with a set of metrics that allow users to evaluate these predictions. By using phylogenetics at its core, AvP represents a substantial improvement compared to existing best BLAST hit-based approaches. Moreover, the metrics they develop help to address many of the issues with HGT detection (e.g. contamination in assemblies and databases). I was able to install the software from GitHub and the documentation is clear. I have no doubt that AvP will be well used and well cited by researchers interested in HGT. I struggled to find anything to complain about and, as such, I would be happy to recommend the manuscript once a few of my very minor comments (below) have been addressed/considered.

1. Lines 32-38: It took me a few readings of this paragraph to work out that the authors were suggesting that the existing methods that reconcile gene/species trees were not ideal because of the requirement of a species tree - could it be reworked slightly to make the message clearer? Even adding “However” before “To achieve this” would help. Note there are a few typos in here too: hundred > hundreds, compare > comparing, wth > with.

2. Lines 57-59: “AI features file can be automatically generated with the Alienness webserver” - I note that this is scored out in the GitHub instructions; is this no longer true?

3. Lines 91-92: “is all not in Ingroup” > “is all species not in Ingroup”

4. Lines 153-154: note that the sequenced Sciento strain of tardigrades was redescribed as H. exemplaris. I’ll let the authors decide if they'd rather use H. dujardini for consistency with previous publications or H. exemplaris for taxonomic correctness.

5. Lines 185-204: The authors only discuss the two problematic cases in their run on C. elegans, which is valuable, but what about all the other predicted HGT loci? How many were there? Were any of interest or consistent with previous work? I understand that analysing this in-depth is beyond the remit of this manuscript, but it seems odd to not mention anything other than where this had problems. A few examples of what can be discovered with the pipeline would be a valuable addition.

6. Lines 201-203: “Consequently, it seems that this new AHS metric is able to correct errors due to contamination and taxonomic assignment bias.” Perhaps I’m confused but doesn’t the AHS value, which suggests that the rodent protein is a strong HGT candidate (10356), suggests the opposite? Surely AHS is positively misleading in this case, and even more so than AI? Or are the authors suggesting that it’s the difference between AHS and AI that is indicative of a false positive?

7. Github: it would be great if the authors could add some worked examples (e.g. the tardigrade and/or C. elegans examples in the paper) to show users how to set this up.

Reviewer #2: I appreciate that Koutsovoulos et al. put multiple steps that are used to detect HGT into one automatic tool called “AvP”. The study is written well and the tool is friendly to users.

I have four main concerns that authors might wish to address.

1. The AvP incorporated the AI values and phylogenetic robustness. Not sure how different AI cutoff will influence outputs. Of courses, there is no standard criterion. Given the complexity of the database, sometimes, the database could contain contamination so that the query protein is highly similar to the hit from contamination. A recent paper (https://pubmed.ncbi.nlm.nih.gov/35853453/) has considered anther parameter “outg_pct”, that is the percentage of species from OUTGROUP lineage in the list of the top 1,000 hits that have different taxonomic species names. Did you consider this too? If not, why or discuss it?

2. The AvP is aiming to detect HGT based on protein-coding genes (or protein sequence), which has been done many previous studies (although they built their own pipelines). Would you please extend your tool to non-coding sequences? If you can make it, that will be much helpful, because nearly all published studies have not done this. And I think non-coding sequences could aslo be HGT too.

3. It seems that the graph neural network (or machine learning) was used to predict the HGT. if it is possible, please compare your tool with it in term of accuracy of HGT identification. If this’s hard, it would be good to discuss this aspect in your study for future direction.

4. If The AvP tool can also report the characteristics of HGT including the codon usage bias, dn/ds, the sequence similarity between donor and recipient, which would be better.

**Have the authors made all data and (if applicable) computational code underlying the findings in their manuscript fully available?**

Reviewer #1: Yes

Reviewer #2: Yes

PLOS authors have the option to publish the peer review history of their article (what does this mean?). If published, this will include your full peer review and any attached files.

Reviewer #1: No

Reviewer #2: No
---

## [Editor Report · Decision Letter 1]

17 Oct 2022

Dear Dr Koutsovoulos,

Thank you for the revised manuscript which I see has addressed most of the reviewer comments.

For your general information, if you agree with the reviewers' comments, it is good form to address them in the manuscript, as it is likely that the readership will have similar concerns. In the rebuttal letter, indicate which lines of the manuscript have been added/amended. This is relevant to points R1P2, R2P1, R2P2. If you could please amend the manuscript and rebuttal letter to this effect it can be send to reviewer 2 again.

Regards,

Mark Ziemann, PhD

---------------

We cannot make any decision about publication until we have seen the revised manuscript and your response to the reviewers' comments. Your revised manuscript is also likely to be sent to reviewers for further evaluation.

Sincerely,

Mark Ziemann

Academic Editor

PLOS Computational Biology

William Noble

Section Editor

PLOS Computational Biology

Dear Dr Koutsovoulos,

Thank you for the revised manuscript which I see has addressed most of the reviewer comments.

For your general information, if you agree with the reviewers' comments, it is good form to address them in the manuscript, as it is likely that the readership will have similar concerns. In the rebuttal letter, indicate which lines of the manuscript have been added/amended. This is relevant to points R1P2, R2P1, R2P2. If you could please amend the manuscript and rebuttal letter to this effect it can be send to reviewer 2 again.

Regards,

Mark Ziemann, PhD
---

## [Decision Letter · Decision Letter 2]

26 Oct 2022

Dear Koutsovoulos,

We are pleased to inform you that your manuscript 'AvP: a software package for automatic phylogenetic detection of candidate horizontal gene transfers.' has been provisionally accepted for publication in PLOS Computational Biology.

Best regards,

Mark Ziemann

Academic Editor

PLOS Computational Biology

William Noble

Section Editor

PLOS Computational Biology

Reviewer's Responses to Questions

**Comments to the Authors:**

Reviewer #2: Thank you very much for including my suggestions! I don’t have any concerns and think it’s ready for publication.

**Have the authors made all data and (if applicable) computational code underlying the findings in their manuscript fully available?**

Reviewer #2: Yes

PLOS authors have the option to publish the peer review history of their article (what does this mean?). If published, this will include your full peer review and any attached files.

Reviewer #2: **Yes: **Xing-Xing Shen

---

## [Editor Report · Acceptance letter]

4 Nov 2022

PCOMPBIOL-D-22-01025R2 

AvP: a software package for automatic phylogenetic detection of candidate horizontal gene transfers.

Dear Dr Koutsovoulos,

I am pleased to inform you that your manuscript has been formally accepted for publication in PLOS Computational Biology. Your manuscript is now with our production department and you will be notified of the publication date in due course.

With kind regards,

Zsofia Freund
